# SINKHORN OUTPUT PERTURBATIONS: STRUCTURED PSEUDO-LABEL NOISE IN SEMI-SUPERVISED SEGMENTATION

## ABSTRACT

In semi-supervised segmentation, the strong-weak augmentation scheme has gained significant traction. Typically, a teacher model predicts a pseudo-label or consistency target from a weakly augmented image, while the student is tasked with matching the prediction when given a strong augmentation. However, this approach, popularized in self-supervised learning, is constrained by the model's current state. Even though the approach has led to state-of-the-art improvements as part of various algorithms, the inherent limitation, being confined to what the teacher model can predict, remains.

In Sinkhorn Output Perturbations, we introduce an algorithm that adds structured pseudo-label noise to the training, extending the strong-weak scheme to perturbations of the output beyond just input and feature perturbations. Our strategy softens the inherent limitations of the student-teacher methodologies by constructing noisy yet plausible pseudo-labels. Sinkhorn Output Perturbations impose no specific architectural requirements and can be integrated into any segmentation model and combined with other semi-supervised strategies. Our method achieves state-of-the-art results on Cityscapes and presents competitive performance on Pascal VOC 2012, further improved upon combining our with another recent algorithm. The experiments also show the efficacy of the reallocation algorithm and provide further empirical insights into pseudo-label noise in semi-supervised segmentation. Code is available at:

## 1 INTRODUCTION

Semi-supervised segmentation algorithms typically focus on two main areas. Most models adopt a student-teacher paradigm. The teacher can be realized in various forms, ranging from the same model as the student as an exponential moving average of the student's weights (Tarvainen & Valpola, 2017), incorporating estimated batch statistics (Cai et al., 2021) or as an independent model and architecture (Chen et al., 2021; Mendel et al., 2020). The teacher model generates targets for the student model, formulated either as a pseudo-labeling (Chen et al., 2021; Hu et al., 2021; Yang et al., 2022) or consistency regularization task (French et al., 2020; Xie et al., 2020). Another area that has been extensively explored is the variations between the inputs of the student and teacher models. The strong-weak augmentation schemes are a key component in successful self-supervised representation learning (Grill et al., 2020; Caron et al., 2021; Chen et al., 2020). The perturbations range from crops or color jitter in the image space to features space methods like dropout (Yang et al., 2023) or adversarial noise (Miyato et al., 2018).

Although self- and semi-supervised approaches have improved substantially, bootstrapping the learned representations solely from the model's current state presents an inherent limitation. These approaches will concentrate on representations that are already encoded in the model's state and can appear under various amounts of input or feature augmentations. Learning to detect an object that is never recognized independent of the augmentations is not possible when the perturbations are limited to the input space. By depending on input and feature space variations and using a closely related teacher model, there is a risk of merely amplifying the existing training signal. Attempting to improve the current model without an external signal places a potential limit on the achievable performance.

In Sinkhorn Output Perturbations (SOP), we investigate the effects of shifting the strong-weak augmentation perspective, adding the external signal as perturbations of the model's outputs. Our objective is to introduce potential targets for the student model that are not covered by the current parameters and thus guide the parameter update away from amplifying the current signal. This pseudo-label noise, distinct from the inherent label noise of an imperfect teacher, is the core of SOP. This contribution is made possible by:

- our proposed interpolation scheme that searches the model's predictions for evidence and constructs plausible alternative label distributions;
- the formulation of output perturbation as an optimal transport problem, enabling accurate and efficient reallocation of pseudo-labels during training;
- an extensive experimental section showing the influence of each hyperparameter and giving further insight into the accuracy of our algorithm compared to and in combination with the state-of-the-art.

## 2 RELATED WORK

Many semi-supervised segmentation approaches have started to include CutMix (Yun et al., 2019) as a critical component in the general scheme. Hu et al. (2021); Zhao et al. (2023b) guide the mixing to promote difficult or rare labels. Other approaches decouple what information is shared between student and teaching models (Jin et al., 2022) add auxiliary teachers (Liu et al., 2022) or completely separate the two (Chen et al., 2021). Yang et al. (2023) extends Sohn et al. (2020) to semi-supervised segmentation and incorporates perturbations in the image and feature spaces. Wang et al. (2022) specifically targets the reliability of the pseudo-labels, and Zou et al. (2021) fuses predictions and Grad-Cam output for refined labels. Arazo et al. (2020) investigate the confirmation bias that pseudo-labeling approaches are prone to, and Zhao et al. (2023a) incorporates the quantitative hardness of the sample to control its influence. He et al. (2021) uses the class distributions in the labeled subset to force the pseudo-label distribution to show similar characteristics and Yuan et al. (2021) investigate distribution mismatches caused by strong augmentations. Semi-supervised segmentation is a broad field and Peláez-Vegas et al. (2023) give a comprehensive view over many competing ideas. Optimal transport has been used in semi and self-supervised settings to force a uniform distribution of the representations (Caron et al., 2020) or to adjust the class distribution of the unlabeled subset according to what is known from the labeled data (Tai et al., 2021; Nguyen et al., 2023).

## 3 SINKHORN OUTPUT PERTUBATION

Sinkhorn Output Perturbations (SOPs) extend a semi-supervised approach where a mean-teacher model generates pseudo-labels for the student network. The algorithm incorporates input perturbations, such as color augmentations and CutMix (Yun et al., 2019; French et al., 2020), aligning with the current state-of-the-art. The novel contribution of SOP is the perturbation at the prediction level. This step involves identifying a perturbed yet plausible target distribution and subsequently reallocating predictions to match this target. The following sections will first introduce the target perturbation scheme, detail the derivation of the allocation algorithm, and outline the full semi-supervised training objective.

### 3.1 NOTATION

Let $\mathbf{x}$ represent an image, and $\mathbf{y}$ denote its corresponding label map with depth $c$, with a height and width $h$ and $w$, respectively. For unlabeled data $\mathbf{x}^u$, the softmax ($sm$) probabilities of the student network $S(\mathbf{x}^u)$ are given by $\boldsymbol{y} \in \mathbb{R}^{c \times h \times w}$. The corresponding predictions from the teacher model are represented by $\hat{\boldsymbol{y}} \in \mathbb{R}^{c \times h \times w}$. The teacher model is derived from the exponential moving average of the student's parameters: $\theta_t^{l+1} = \omega \theta_t^l + (1 - \omega)\theta_s^l$. Here, $\theta_s^l$ and $\theta_t^l$ refer to the student's and teacher's parameters at iteration $l$. $\mathbf{1}_d$ is the $d$-dimensional vector containing only 1s and similar to Tai et al. (2021) we define $x^+ := \max(0, x)$. For simplicity, the predictions are flattened from $\mathbb{R}^{c \times h \times w}$ to $\mathbb{R}^{d \times c}$, with $d = hw$ representing the flattened dimension. Iterating over the $d$ entries corresponds to iterating over the height and width of the prediction.

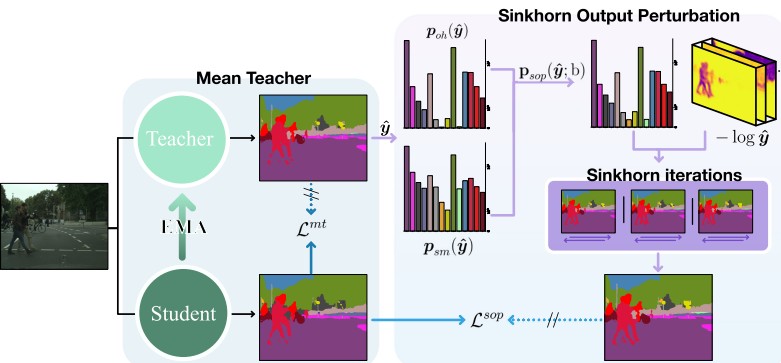

Figure 1: **An unsupervised iteration with Sinkhorn Output Perturbations -** SOP extends semi-supervised segmentation with perturbations in the output space. The prediction of the teacher model is used to calculate the distributions $\boldsymbol{p}_{oh}$ and $\boldsymbol{p}_{sm}$. Then, the Sinkhorn-Knopp algorithm optimizes the output to resemble $\mathbf{p}_{sop}$. Both the predicted and perturbed pseudo labels are used to optimize the student on unlabeled images.

## 3.2 PERTURBATION SCHEME

SOPs consist of finding a perturbed target distribution and then reallocating the predictions to match this target. Our approach consists of three steps.

We extract information from the full depth of each prediction and look further than only the largest probabilities. Then, this newly found evidence is merged with the distribution of the pseudo-label frequency controlled by a random factor. Finally, the algorithm subdivides this process into non-overlapping views to increase the per-sample variety of the new targets.

### 3.2.1 SOFTMAX-MINING

State-of-the-art approaches in image classification evaluated on the ImageNet dataset achieve top-1 accuracies surpassing 90%. For the top-1 accuracy, only the prediction with the highest probability is relevant, leading to a one-hot (*oh*) representation of the output. We define the one-hot encoding $\mathbf{I}_y$ of a probability vector $y \in \mathbb{R}^c$ as:

$$\mathbf{I}_y = \begin{cases} 1 & \text{if } y_j = \max(y), \ j \in c \\ 0 & \text{otherwise,} \end{cases} \quad (1) \qquad \mathbf{A}_y = \begin{cases} 1 & \text{if } y_j \neq 0, \ j \in c \\ 0 & \text{otherwise.} \end{cases} \quad (2)$$

The presence encoding $\mathbf{A}_y$ is a binary vector that indicates which elements of a vector $y$ are non-zero. To compute the label distribution $\boldsymbol{p}_{oh}(\boldsymbol{y}) \in \mathbb{R}^c$ based on pixel-wise one-hot encoding over the entire prediction $\boldsymbol{y} \in \mathbb{R}^{d \times c}$ is given by:

$$\boldsymbol{p}_{oh}(\boldsymbol{y}) = \frac{1}{d} \sum_i^d \mathbf{I}_{\boldsymbol{y}_i}. \quad (3)$$

The resulting *oh*-distribution focuses only on predictions with the highest probability and does not capture the ranking of output probabilities.

Prior to error rates of less than 10%, the top-5 performance served as a commonly used metric to compare architectures on ImageNet. This metric captures if the correct class is among the five predictions with the highest probability. In general, top-5 accuracies were higher than top-1 results because the model essentially gets multiple chances to identify the correct class. This emphasizes that the order and magnitude of all predictions contains information that is not captured by just a one-hot encoded representation of the model's output. With softmax-mining, SOP accounts for this property by computing the element-wise sum of prediction vectors:

$$s(\boldsymbol{y}) = \mathbf{A}_{\boldsymbol{p}_{oh}(\boldsymbol{y})} \sum_i^d \boldsymbol{y}_i \quad (4) \qquad \boldsymbol{p}_{sm}(\boldsymbol{y}) = \frac{s(\boldsymbol{y})}{\|s(\boldsymbol{y})\|_1}. \quad (5)$$

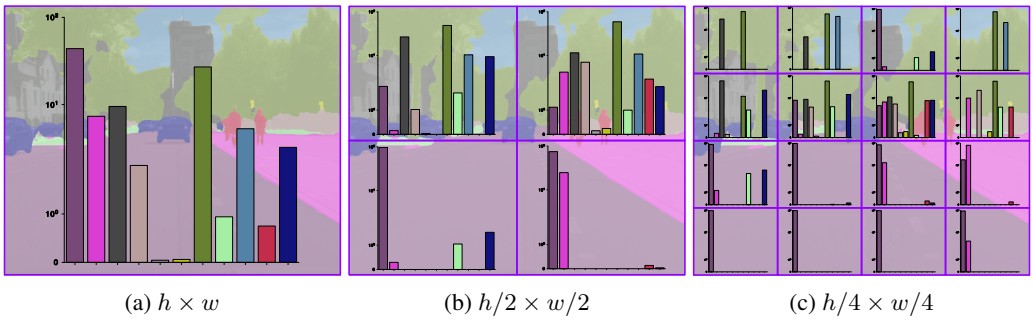

(a) $h \times w$        (b) $h/2 \times w/2$        (c) $h/4 \times w/4$

Figure 2: The **patch size** controls the local versus global tradeoff for the target distributions.

The *sm*-distribution $\boldsymbol{p}_{sm}(\boldsymbol{y}) \in \mathbb{R}^c$ is described as the normalized, filtered element-wise vector sum of the probability vectors $\boldsymbol{y}_i$. Filtering by $\mathbf{A}_{\boldsymbol{p}_{oh}(\boldsymbol{y})}$ removes the classes that are absent from the one-hot distribution $\boldsymbol{p}_{oh}(\boldsymbol{y})$ and were never among the output classes with the highest probability. The probabilities of the present classes accumulate across all spatial locations, shifting the *sm*-distribution compared to the one-hot encoded outcome.

The intent of softmax-mining is to accumulate evidence. If a class frequently ranks just below the largest probabilities, $\boldsymbol{p}_{sm}$ will reflect that.

### 3.2.2 INPUT PATCHING

The previous section discussed approaches to turn segmentations into label distributions, but a single, global label distribution of a segmentation can obscure the intricate location-specific details. Fragmenting the prediction into patches and calculating the distribution per patch rather than per image, the context window for the distribution is decreased, thereby emphasizing local characteristics. At the *pixel level*, segmentation can be interpreted as $hw$ distributions for patches with a size of $1 \times 1$, providing exact local information. Modulating between a patch size of $1 \times 1$ and $h \times w$ grants control over the local-global interaction and adjusts how large the context window is for each distribution. Figure 2 gives an overview of different patch sizes and their influence on the resulting distributions.

Predictions can be flattened from a batch representation $\boldsymbol{y} \in \mathbb{R}^{b \times c \times h \times w}$ into the patched representations, denoted by $\mathbf{x}_p \in \mathbb{R}^{n \times c \times p_h \times p_w}$. This gives the new batch size $n = \frac{bhw}{p_h p_w}$ with $h$ and $w$ as the original height and width and the patch sizes $p_h$ and $p_w$.

Given the non-uniform distribution of objects across the image, limiting the context window will result in significant differences between distributions at varying patch sizes. Beyond the local-global tradeoff of the captured distribution, more patches introduce a mechanism to diversify the reallocation targets in SOP - from one per image to one per patch.

### 3.2.3 BETA - INTERPOLATION

The previous sections introduced two distinct class distributions and described patch-based sub-sampling in SOP. One distribution directly reflects the content of the pseudo-labels, and the other distribution emphasizes the ranking information present in the outputs. To balance both perspectives, in SOP both distributions are merged in an interpolation step in order to create the perturbed target distribution.

The interpolation weight b determines the emphasis on the mined evidence. b is derived from a Beta distribution[1], modulated by the parameter $\phi$:

$$\mathbf{b} = 2\phi\psi - \phi, \psi \sim Beta(\alpha, \beta), \phi \in [0, 1]. \tag{6}$$

When $\alpha = \beta = 0.5$, the values produced by equation 6 primarily cluster around the boundaries, ranging between $\phi$ and $-\phi$.

---

[1]Visualization of the pdf and cdf are given in Appendix A.5

(a) $\mathbf{p}_{oh}(\hat{\boldsymbol{y}})$      (b) $\mathbf{p}_{sop}(\hat{\boldsymbol{y}}; 0.05)$      (c) $\mathbf{p}_{sop}(\hat{\boldsymbol{y}}; -0.05)$

Figure 3: **The influence** the interpolation parameter **b** has on the target distribution. A small positive interpolation value will in general allocate more towards the rare classes like *traffic lights, car* or *bicycle*. A negative value will remove underrepresented classes, based on the result of the softmax-mining.

Direct interpolation between the predicted and mined distributions, controlled by $\psi$, would predominantly select either the one-hot or the mined structure, offering limited intermediate choices. The hyperparameter $\phi$ narrows this effect. This scaling method explicitly permits negative values.

Using a negative b counters the evidence in the prediction and effectively removes low-frequency classes:

$$t_{sop}(\hat{\boldsymbol{y}}; \mathrm{b}) = (1 - \mathrm{b})\boldsymbol{p}_{oh}(\hat{\boldsymbol{y}}) + \mathrm{b}\boldsymbol{p}_{sm}(\hat{\boldsymbol{y}}), \quad (7) \qquad \mathbf{p}_{sop}(\hat{\boldsymbol{y}}; \mathrm{b}) = \frac{t_{sop}(\hat{\boldsymbol{y}}; \mathrm{b})^+}{\|t_{sop}(\hat{\boldsymbol{y}}; \mathrm{b})^+\|_1}. \qquad (8)$$

To ensure that $\mathbf{p}_{sop}(\cdot)$ remains a valid distribution, the negative values in the interpolation are removed, and the result is normalized.

Figure 3 shows the one-hot distribution for a predicted segmentation and the interpolated and perturbed views of this output. Allowing to move away from the mined distribution will remove low-frequency classes and prohibit continuous region growth. Interpolation only in the direction of the mined distribution can cause the model to exaggerate boundaries around identified objects. In SOPs, the explicit inclusion of negative interpolation weights and patch-based sub-sampling will result in multiple allocation trajectories and introduce new diversity to the pseudo-labels.

### 3.3 BATCHED SINKHORN-KNOPP ALGORITHM

In the SOP framework, the cross-entropy $H(\cdot, \cdot)$ is utilized to compare the student's prediction against both the ground truth for labeled data and the teacher's pseudo-labels for unlabeled data. The cross-entropy also plays a guiding role in shaping the perturbed labels.

Sections 3.2.1 and 3.2.3 elaborate on the derivation of the target distribution for output perturbation from the model's prediction. To extract a flattened segmentation $\boldsymbol{y}^\star$ that satisfies the condition $\mathbf{p}_{oh}(\boldsymbol{y}^\star) = \mathbf{p}_{sop}(\hat{\boldsymbol{y}}, \mathbf{b})$, a linear program is constructed based on the cross-entropy between the perturbed segmentations $\boldsymbol{y}^\star$ and predicted segmentations $\hat{\boldsymbol{y}}$:

$$\min_{\boldsymbol{y}^\star \in \mathcal{P}} \sum_{i}^{d} H(\boldsymbol{y}_i^\star, \hat{\boldsymbol{y}}_i) = \min_{\boldsymbol{y}^\star \in \mathcal{P}} \langle \boldsymbol{y}^\star, \mathbf{C} \rangle \qquad (9)$$

Here, $\mathbf{C}$ is defined as $-\log \hat{\boldsymbol{y}}$. The constraints for rows and columns are as follows:

$$\mathcal{P} = \left\{ \boldsymbol{y}^\star \in \mathbb{R}_+^{d \times c} \mid \boldsymbol{y}^\star \mathbf{1}_c = \frac{1}{d}\mathbf{1}_d \text{ and } \boldsymbol{y}^{\star T}\mathbf{1}_d = \mathbf{p}_{sop}(\hat{\boldsymbol{y}}; \mathbf{b}) \right\}. \qquad (10)$$

Constraining the uniformity of the rows and directing the column constraint to the desired target distribution, equations 9 and 10 formulate an optimal transport problem. By introducing an entropy constraint, $\epsilon H(\boldsymbol{y}^\star)$ to equation 9 controlling the smoothness of the solution, the problem can be solved efficiently by the Sinkhorn-Knopp algorithm (Cuturi, 2013).

The Sinkhorn algorithm operates as an iterative method for column and row normalization, progressively refining scaling vectors $f, g$ over the iterations:

$$f_i^{(l+1)} = \epsilon \log \frac{1}{d} - \epsilon \log \sum_{k}^{d} e^{-\mathbf{C}_{ki}/\epsilon} e^{g_i^{(l)}/\epsilon} , \ g_i^{(l+1)} = \epsilon \log \mathbf{p}_{sop}(\hat{\boldsymbol{y}}) - \epsilon \log \sum_{k}^{c} e^{-\mathbf{C}_{ik}/\epsilon} e^{f_i^{(l+1)}/\epsilon}.$$
$$(11)$$

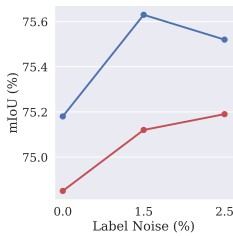 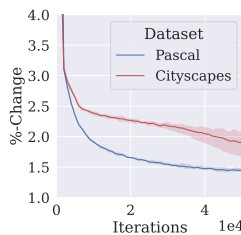 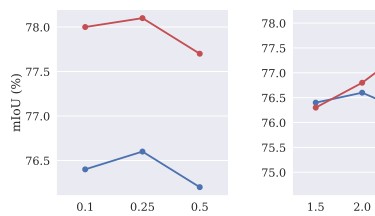

Figure 4: **Adding random label-noise to the pseudo-labels** on the 12.5% partition.

Figure 5: **Average number of changed labels** by SOP over the three partitions.

Figure 6: **Influence of the trade-off weight $\lambda$ and the perturbation weight $\tau$** on both Pascal and Cityscapes with 12.5% of the labels.

Expressed in $\log$-space, the optimal solution to the linear program in 9 is:

$$\boldsymbol{y}_{ij}^{\star} = e^{f_i/\epsilon} e^{-\mathbf{C}_{ij}/\epsilon} e^{g_j/\epsilon} \tag{12}$$

A key advantage of the Sinkhorn algorithm is its inherent parallelism. It allows for the concurrent computation of multiple assignments on a GPU, which is consistent with the SOP's objective of perturbing batches of predictions with unique target distributions. A Pytorch style implementation is given in Appendix A.1.

### 3.4 TRAINING OBJECTIVE

Given the perturbed label, the resulting model produces three outputs for an unlabeled image: the student's prediction $\boldsymbol{y}$ from an augmented view $\mathbf{x}^u$, the teacher model prediction $\hat{\boldsymbol{y}}$ and their perturbations $\boldsymbol{y}^{\star}$. To compute the final loss, we utilize a layered weighting of these individual components. The mean-teacher and SOP losses adopt a confidence weighting where the largest probability of the teaching model scales the unsupervised loss, as suggested in Mendel et al. (2020):

$$\mathcal{L}_i^{mt} = (\max \hat{\boldsymbol{y}}_i)^\gamma H(\mathbf{I}_{\hat{\boldsymbol{y}}_i}, \boldsymbol{y}_i) \text{ and } \mathcal{L}_i^{sop} = (\max \hat{\boldsymbol{y}}_i)^\gamma H(\mathbf{I}_{\boldsymbol{y}_i^{\star}}, \boldsymbol{y}_i) \tag{13}$$

The overall objective for a single input consists of the cross-entropy between the student's prediction on labeled data combined with the weighted sum of unsupervised results:

$$\mathcal{L} = \frac{1}{d} \sum_i^d (H(\mathbf{y}_i, S(\mathbf{x})_i) + \lambda(\mathcal{L}_i^{mt} + \tau \mathcal{L}_i^{sop})) \tag{14}$$

The trade-off weight $\lambda$, inspired by Chen et al. (2021), modulates the impact of the semi-supervised component. Additionally, we introduce a separate weighting $\tau$ that exclusively scales the influence of the perturbed outcomes. This layered approach empowers the model to place greater emphasis on the semi-supervised objective while maintaining control over the influence of perturbations. The three-tiered weighting mechanism dynamically differentiates between high and low-confidence predictions and distinctly separates the perturbations from the teacher's predictions.

## 4 EXPERIMENTS

The section covers the main comparison of SOP with recent semi-supervised approaches with limited labeled data. Further, we provide an analysis of the effects of just label noise. An ablation study covering the main hyperparameters concludes the section.

### 4.1 RANDOM PERTURBATIONS

As a baseline, we show the effects of added unstructured random noise to the pseudo-labels. To add bits of label noise, the predictions are downsampled by a factor of 0.25, and at random locations, the predicted label is replaced by a randomly selected one. The subsequent upsampling to the original size increases each reallocation's area while preserving the total ratio.

Table 1: **Comparison with state-of-the-art** on the Cityscapes validation sets. All methods are based on a DeepLabv3+.

| Method | ResNet50 | | | ResNet101 | | |
|---|---|---|---|---|---|---|
| | 6.25% | 12.5% | 25% | 6.25% | 12.5% | 25% |
| CPS | 74.4 | 76.6 | 77.8 | 74.7 | 77.6 | 79.2 |
| AEL | - | - | - | 75.8 | 77.9 | 79.0 |
| ST++ | - | 72.7 | 73.8 | - | - | - |
| PS-MT | - | 77.1 | 78.3 | - | - | - |
| U$^2$PL | - | - | - | 70.3 | 74.3 | 76.4 |
| iMAS | 74.3 | 77.4 | 78.1 | - | - | - |
| UniMatch | 75.0 | 76.8 | 77.5 | 76.6 | 77.9 | 79.2 |
| AugSeg | 73.7 | 76.4 | **78.7** | 75.2 | 77.8 | **79.5** |
| **SOP (Ours)** | **76.0** | **78.1** | 78.4 | **77.6** | **78.7** | **79.5** |

Figure 4 shows the change in mIoU with increased label noise. On Cityscapes, changing 2.5% of the predictions to random labels improves the mIoU semi-supervised baseline without SOP from 74.8 to 75.1. Similarly, on Pascal VOC 2012, both amounts of added noise improve the validation results from 75.1 to 75.6 and 75.5. Figure 5 plots the average number of reallocated labels by SOP over the course of training. As the model converges, the predictions $\hat{y}$ become sharper, resulting in both less evidence for the auxiliary classes and a finer cost map. This, for the selected hyperparameters, leads to SOP almost mirroring the optimal noise ratios from the randomized experiments. Adding random noise to the pseudo labels enforces a fixed floor for the difference between the prediction and the targets and thus, we hypothesize, encourages the model to continue to learn.

## 4.2 MAIN RESULTS

The following section compares SOP on various data partitions with a set of published results. We have verified that our experiments are run with the same labeled - unlabeled splits as the published algorithms and also employ the same pretrained weights as UniMatch (Yang et al., 2023) and AugSeg (Zhao et al., 2023b).

### 4.2.1 CITYSCAPES

Table 1 shows the mIoU of SOP on the 6.25%, 12.5% and 25% partitions with both ResNet50 and ResNet101. SOP consistently either outperforms or is close to the published competition. Especially on the 6.25% and 12.5% partitions, SOP is able to improve the mIoU by 1.0 and 0.7 for the ResNet50 backbone and 1.0 and 0.8 for the ResNet101 backbone in comparison to the competition. On the 25% partition SOP did not match AugSeg's (Zhao et al., 2023b) performance with the ResNet50 backbone but could achieve parity with the ResNet101 backbone.

### 4.2.2 PASCAL

On Pascal VOC 2012, we provide results on the blended dataset where the labeled samples are from both the regular Pascal VOC 2012 where the labeled data is only from the high-quality set (Table 2, right) or also from the extended SBD dataset (Table 2, left).

With the ResNet50 backbone, SOP is comparable to the recently published methods, with the largest discrepancy being on the 6.25% partition of 2.2 mIoU to the best-performing model. On the 25% partition with a ResNet101 backbone, SOP achieves the second best result with a mIoU of 79.0, a 0.2 difference to UniMatch (Yang et al., 2023). SOP's results are in the lower third for the remaining partitions. A second evaluation is training the models with just access to the high-quality samples, and is given in Table 2 on the right. Here, SOP achieves results in the upper third, with only the most recent algorithms outperforming SOP's mIoU.

## 4.3 COMPARISON TO UNIMATCH

The main thesis of the paper is that in addition to input and feature perturbation, output perturbation in the form of structured noise can improve the model's performance. The recently proposed Uni-

Table 2: **Comparison with state-of-the-art** on the Pascal VOC 2012 validation sets. The training images are sampeled from the blended dataset (left) or from the original 1464 images (right). All methods are based on a DeepLabv3+ architecture.

| | ResNet50 | | | ResNet101 | | | | ResNet101 | | |
|---|---|---|---|---|---|---|---|---|---|---|
| **Method** | 6.25% | 12.5% | 25% | 6.25% | 12.5% | 25% | **Method** | 92 | 183 | 366 |
| CPS | 71.9 | 73.6 | 74.9 | 74.4 | 76.4 | 77.6 | CPS | 64.0 | 67.4 | 71.7 |
| AEL | - | - | - | 77.2 | 77.5 | 78.0 | U$^2$PL | 67.9 | 69.1 | 73.6 |
| ST++ | 73.2 | 75.5 | 76.0 | 74.7 | 77.9 | 77.9 | ST++ | 65.2 | 71.0 | 74.6 |
| PS-MT | 72.8 | 75.7 | 76.4 | 75.5 | 78.2 | 78.7 | PS-MT | 65.8 | 69.5 | 76.5 |
| iMAS | 74.8 | 76.5 | 77.0 | 76.5 | 77.9 | 78.1 | iMAS | 68.8 | 74.4 | 78.5 |
| UniMatch | **75.8** | **76.9** | 76.8 | **78.1** | **78.4** | **79.2** | UniMatch | **75.2** | **77.2** | **78.8** |
| AugSeg | 74.6 | 75.9 | **77.1** | 77.0 | 77.3 | 78.8 | AugSeg | 71.0 | 75.4 | **78.8** |
| | | | | | | | GTA-Seg | 70.0 | 73.1 | 75.5 |
| **SOP (Ours)** | 73.6 | 76.6 | 76.6 | 75.4 | 77.3 | 79.0 | **SOP (Ours)** | 70.2 | 73.6 | 78.5 |

Table 3: **SOP and Uni-Match** on Pascal. (‡ our reproduction.)

| **Method** | 92 | 183 |
|---|---|---|
| UniMatch | 75.2 | 77.2 |
| UniMatch‡ | 71.6 | 77.5 |
| **+ SOP** | **75.8** | **79.0** |

Table 4: **Patchsize** controlling the local-global context.

| **Patches** | City | Pascal |
|---|---|---|
| $h/1 \times w/1$ | 77.8 | 75.7 |
| $h/2 \times w/2$ | 78.0 | 76.2 |
| $h/4 \times w/4$ | **78.1** | **76.6** |

Table 5: **Varying allocation** values for $\phi$ in Eq. (6)

| $\phi$ | City | Pascal |
|---|---|---|
| 0.2 | 77.7 | 75.8 |
| 0.3 | 77.9 | 76.0 |
| 0.4 | **78.1** | **76.6** |

Table 6: **Varying allocation** values for $\alpha = \beta$ in Eq. (6)

| $\alpha, \beta$ | City | Pascal |
|---|---|---|
| 0.1 | **78.1** | 76.1 |
| 1.0 | 77.6 | 75.9 |
| 0.5 | **78.1** | **76.6** |

Match algorithm heavily leans on the input perturbation component. To show that UniMatch can further benefit from added output perturbation, we integrate SOP with their published training code and run experiments on Pascal VOC 2012. We reproduce results with the base UniMatch algorithm and implement a version with the setup from equation 13 and extend each of the three pseudo-label objectives in UniMatch with the perturbed labels.

Table 3 shows the results on the original Pascal 2012 set for 92 and 183 labels. On the 92 label partition, the extended UniMatch achieves a mIoU of 75.8 and improves upon the published result, although our initial reproduction falls short, with a mIoU of 71.6 compared to 75.2. On the 183 partition, our reproduction matched the published results. Further, in this case, the addition of SOP improves the mIoU by 0.8 to 79.0.

### 4.4 ABLATION

The ablation covers the effect of the hyperparameters introduced in Sections 3.2.3, 3.2.2 and 3.4 on the 12.5 % partition for a model with the ResNet50 backbone.

#### 4.4.1 INFLUENCE OF INPUT-PATCH GRANULARITY

The results in Section 4.2 all employed a patch size of $h/4 \times w/4$, see Figure 2 for reference. The size controls the local to global influence, with a larger number of small patches leading to perturbations based on more localized information. Table 4 shows the effects of larger patch sizes on Cityscapes and Pascal VOC 2012 for the 12.5% partition. The patch size has a significant effect on the mIoU, with the smallest patch size resulting in the best results. A $h \times w$ patch size, including all global information into the target distribution results in the worst performance of 77.8 on Cityscapes and 75.7 on Pascal VOC 2012. The intermediate configuration also results in an intermediate performance, with a 0.1 difference on Cityscapes and a 0.4 difference on Pascal VOC 2012.

#### 4.4.2 VARYING DEGREE OF ALLOCATIONS

Apart from the patch size, the beta distribution and its scaling are the determining factors in obtaining the SOP target distribution. Tables 5 and 6 highlight the impact of both parameters. The parameter $\phi$

in equation 6 is an upper limit on the difference between the target and the one-hot distribution, with lower values resulting in equation 7 being closer to $\mathbf{p}_{oh}(\hat{\boldsymbol{y}})$. As a result, the diversity of the perturbed pseudo-labels is reduced, thereby diminishing SOPs effect. Table 5 underlines this theory, with $\phi$ of $0.2$ and $0.3$ resulting in a worse performance than $0.4$ on both datasets.

The second critical parameter is the beta distribution, controlled by $\alpha$ and $\beta$. It influences the direction and magnitude in equation 6. The given parameter results in the shape of the distribution transitioning from uniform to step-like, concentrating at the edges (see A.5). With $\alpha = \beta = 0.1$, the sampled values concentrate towards $-\phi$ or $\phi$ with minimal intermediate values. In a uniform setting, all values within the range are equally probable. Table 6 evaluates the performance across the configurations. The uniform setting decreases the performance to 77.6 and 75.9 on Cityscapes and Pascal VOC 2012 respectively. Compared to the other configurations, the uniform allocation will result in more perturbations that are close to $\boldsymbol{p}_{oh}$. As shown with the parameter $\phi$, less diversity limits the performance of SOP. The more concentrated distribution with $\alpha = \beta = 0.1$ also does not improve upon $\alpha = \beta = 0.5$, with a mIoU of 78.1 and 76.1. Again, the diversity can possibly explain the differences since with the step-like cdf, b will be very close to $\{-\phi, \phi\}$ with $\alpha = \beta = 0.1$ and show fewer distinct perturbations.

### 4.4.3 Trade-off and perturbation weights

While factors such as the number of patches and the interpolation weight affect the content of the perturbed pseudo-labels, the parameters $\tau$ and $\lambda$ primarily determine the influence of these labels on the semi-supervised and the overall objectives. Figure 6 presents the outcomes across a range of parameter settings. In their work, Chen et al. (2021) chose a $\lambda$ value of $1.5$ and $6$ for Pascal VOC 2012 and Cityscapes, respectively, as weighting on their semi-supervised contribution. As observable in (Chen et al., 2021), performance on Cityscapes improves with increasing values of $\lambda$ in SOP as well. The parameter $\tau$ modulates the perturbed labels' contribution. For both datasets, too small or large values resulted in a suboptimal mIoU. Control over how much weight is given to the perturbed pseudo-labels is critical. While they provide an alternate perspective, they may not necessarily be more accurate than the predicted labels.

## 5 Discussion

Sinkhorn Output Perturbations achieves state-of-the-art results on Cityscapes and presents competitive performance on Pascal VOC 2012. Combined with UniMatch (Yang et al., 2023), it further improves upon previous results. These outcomes indicate that moving beyond merely input perturbations and incorporating structured label noise can enhance semi-supervised algorithms. Softmax-mining and beta interpolation, as utilized within SOP, are far from optimal. Although presented as components of the algorithm in this paper, they are not necessarily the most suitable methods for introducing structured noise into the pseudo-labels. Their primary role is to produce plausible outcomes and demonstrate that perturbations can be advantageous at every stage of the model: input, intermediate, and output. It is conceivable that alternative methods to generate target distributions, either from general dataset statistics or an approach incorporating the current error rate of the model. Approaches that incorporate error statistics from the subset and perturb the target distribution following the knowledge of the model's limitations could also be a viable way to generate targets.

The perturbations in SOP can be inaccurate, potentially more so than the predicted pseudo-labels. Hence, the weighting factors within SOP must be carefully balanced to establish a ceiling on the introduced noise. Especially the interpolation scheme, which allows both positive and negative values to enhance the diversity of perturbations, is crucial. Preliminary experiments showed that by interpolating between the mined and predicted target distributions using only a positive factor, areas representing low-frequency objects expanded within low-cost regions. This phenomenon resulted in contours forming around underrepresented classes, continuously growing their areas.

Even though the balance of the pseudo-labels in our algorithm needs to be carefully considered, SOP has demonstrated state-of-the-art performances. More advanced strategies for generating target distributions could further improve the performance, leading to an increased prevalence of output perturbations in both semi-supervised and potential self-supervised applications.

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

## A    APPENDIX

### A.1    BATCHED SINKHORN ALGORITHM - PYTORCH

---

**Algorithm 1 Batched Sinkhorn Iterations in PyTorch.** This is an implementation of Equation 11. The parameters are reparametized as $p = f/\epsilon$, $q = g/\epsilon$ and $K = -\mathbf{C}/\epsilon$.

---

```python
# K: NxDxC - cost, p, q: NxDx1, NxCx1 - initialized to zero
# loga: NxDx1 - log 1/D, target: NxC, logb: NxCx1 - log target
for i in range(50):
    q = logb - torch.logsumexp(K + p, 1)[:, :, None]
    p = loga - torch.logsumexp(K + q.transpose(1,2), 2)[:, :, None]
    if i % 20 == 0:
        y_star = torch.exp(p + K + q.transpose(1,2))
        alloc = onehot_distribution(y_star)
        if torch.norm(alloc-target, p=1,dim=1).mean() < 0.001:
            break
```

---

Algorithm 1 showcases the PyTorch implementation suitable for a batch size of $n$, flattened images of dimensions $\mathbb{R}^{n \times d \times c}$, and corresponding $n$ perturbed target distributions for each patch with a flattened size of $d = p_h p_w$. Both broadcasting in Pytorch as well as the inherent properties of the Sinkhorn algorithm enable the parallel optimization of the whole batch.

### A.2    IMPLEMENTATION

All experiments in the paper were run with a DeepLabv3+ (Chen et al., 2018) architecture using ResNet50 and ResNet101 (He et al., 2016) backbones and an output stride of 16. The models were optimized with Stochastic Gradient Descent (Kiefer & Wolfowitz, 1952; Robbins & Monro) with an initial learning rate of $0.01$ for the Cityscapes experiments. On Pascal VOC 2012 the initial learning rate of $0.001$ for the backbone was scaled by a factor of 10 for the remaining layers. The respective batch sizes are $8$ and $16$ depending on the dataset. The momentum parameter was set to $0.9$, weight decay to $1e-4$, and we decayed the learning rate with a polynomial schedule: $lr = lr_{\text{initial}} \times \left(1 - \frac{\text{iter}}{\text{maxiter}}\right)^{0.9}$ over 50,000 training iterations. The parameters $\tau$ and $\gamma$ were set to $0.25$ and $2$ for all main experiments, respectively. The EMA momentum parameter $\omega$ was set to $0.999$ as its default value and $\epsilon$ in the Sinkhorn algorithm to $0.05$. $\lambda$ is set to $2$ on Pascal VOC 2012 and $6$ on Cityscapes.

For Cityscapes, we optimize using an online-hard example mining loss for the supervised objective and employ the regular cross-entropy loss for the Pascal VOC 2012 experiments, as in (Chen et al., 2021; Yang et al., 2023). We adopt the augmentation techniques from UniMatch, beginning with an initial random resizing between $0.5$ and $2.0$ of the original scale, followed by random cropping to resolutions of $800 \times 800$ for Cityscapes and $512 \times 512$ for Pascal. Inputs for the student model are further enhanced with random horizontal flipping, color and grayscale transformations, blurring, and CutMix (Yun et al., 2019) on unlabeled data as proposed in French et al. (2020) and adopted by the state-of-the-art.

### A.3    DISTRIBUTION ORACLE

A core premise of SOP is the plausibility of perturbations. Section 4.1 has shown that just random noise can lead to improvements, but there remains a delta between the performance of random versus guided perturbations. To further verify that the Sinkhorn algorithm and the selected hyperparameters achieve the goal of plausible perturbations, it is vital that the output perturbations match the generated targets and that the changes were made at the most suitable locations. The location of the reassignments is controlled by the cost, the negative log probabilities, in the Sinkhorn iterations.

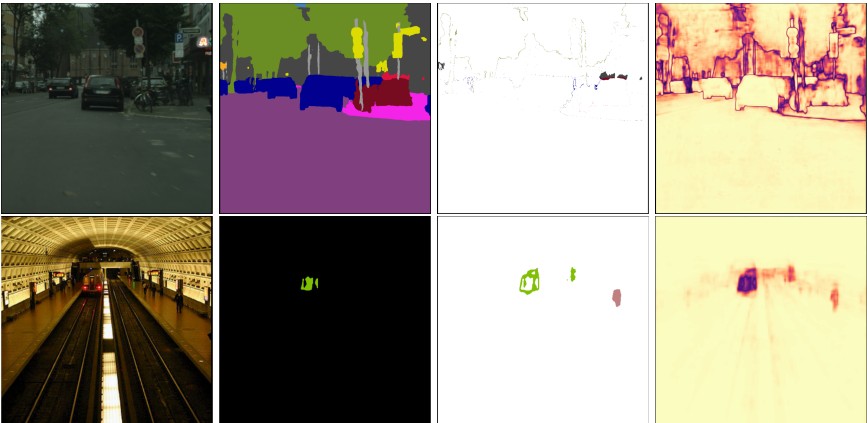

Figure 7: **Sinkhorn Output Perturbation as weakly supervised approach.** With access to the ground truth label distribution, the Sinkhorn algorithm adds or extends the missing classes or removes superfluous predictions. The changes made by the algorithm are shown in color. Unaltered areas are shown in white. The last column displays the largest probability values.

Table 7: **Weakly-supervised** results on the 12.5% partition. The SOP targets are from the labels.

|                           | Cityscapes | Pascal |
|---------------------------|------------|--------|
| **SOP**                   | 78.1       | 76.6   |
| **SOP** + distribution-oracle | **78.8** | **77.5** |

To isolate this component of the algorithm, we design an experiment where the target distribution for the Sinkhorn iterations is obtained from the ground-truth labels. This weakly-supervised approach has global knowledge of the kind of visible objects and their frequencies calculated from the ground truth distribution. So the algorithm only has to assign or rearrange the prediction according to the cost. The experiments were run on the 12.5% partition and a ResNet50 backbone with a patch size of $h/1 \times w/1$ and $\tau = 1$.

Figure 7 shows two images with their predicted segmentations and their reassignments $y^\star$ where the target is calculated from the ground-truth. In the example from Cityscapes, the algorithm removed the initially predicted pedestrian class since it is not present in the ground truth.

The sample from Pascal shows that in a low-cost area, the absent *person* class was added with the Sinkhorn algorithm. The example shows that the allocation cost is the lowest around the general area where the *train* was detected, but the algorithm still found an appropriate region to add the *person*. It, however, also shows that it remains an imperfect adjustment since the added *person* is clustered in one region, and other locations were missed. Similarly, the *train* label was assigned to the wrong areas as well.

Still, Table 7 shows the improvements achievable with access to the ground truth distribution. The fewer per-image objects and the foreground background structure of Pascal VOC 2012 have made knowledge of the ground truth distribution advantageous and improved the mIoU from 76.6 to 77.5. The improvement with weakly supervised SOP is similar to Pascal VOC 2012, even though the performance on the former was already approaching supervised levels with a mIoU of 78.1. The mIoU of 78.8 still represents a significant performance gain. The real-world applications of this specific setup will be limited since the exact knowledge of the kind and size of objects in a scene is information that is likely only available in very few use cases, for example, video segmentation tasks with small intervals of unlabeled frames between labeled key-frames. However, this experiment is not intended to propose an algorithm for weakly labeled segmentation problems but to establish that reallocation with the Sinkhorn algorithm can lead to plausible alternative views from the predicted segmentation.

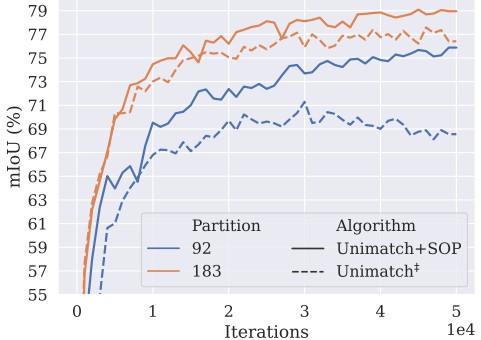 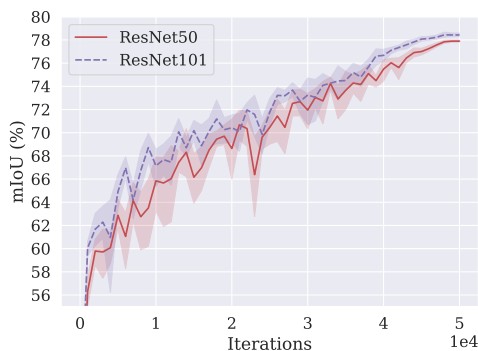

Figure 8: **Comparing UniMatch** with our extended version over the course of training. UniMatch achieved state-of-the-art results on Pascal VOC 2012 and augmented with SOP the results can be improved further.

Figure 9: **Cityscape performance with multiple seeds**. For the $12.5\%$ partition and both ResNet backbones SOP achieves consistent high performance over six independent runs.

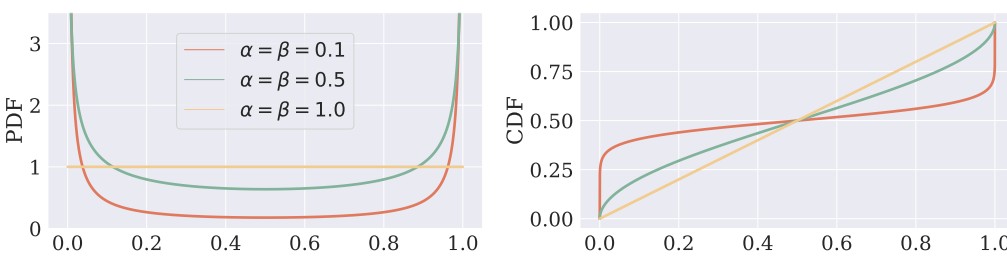

Figure 10: **Properties of the beta distribution**. A value of $0.5$ will result in promoting interpolation between the edges with less frequent intermediate values.

## A.4  TRAINING TRAJECTORIES

Figure 9 plots the validation performance over the course of training for the main result and five additional runs with the same labeled data but with consecutive seeds for the $12.5\%$ partition. SOPs performance is consistent over the multiple runs. Figure 8 compares our reproduced version of UniMatch and our extension. On both partitions, UniMatch+SOP consistently stays above just UniMatch, and in contrast to UniMatch, there is a continuing improvement over the course of training.

## A.5  BETA DISTRIBUTION

The beta distribution plays a key role in controlling the strength of the output perturbations. Figure 10 shows the pdf and cdf of the beta distributions for the selected parameter settings and helps to get an intuition for the range of values b will take during training. The parameter settings used in Table 5 vary between a uniform to a step-like cdf. The best results were achieved with the intermediate configuration.

## A.6  STUDENT-TEACHER AGREEMENT AND ACCURACY

The goal of SOP is to introduce variety into the model and lessen the student-teacher dependency. As shown by the experiments adding just random noise, merely changing the output of the teacher randomly without considering any of its structure or the encoded information and using these as auxiliary targets can improve performance. Figure 11a shows the agreement between the student and teacher models during training, with and without random noise, on Cityscapes for the $12.5\%$ partition. Notably, this is with CutMix augmentations (French et al., 2020), a strong form of input

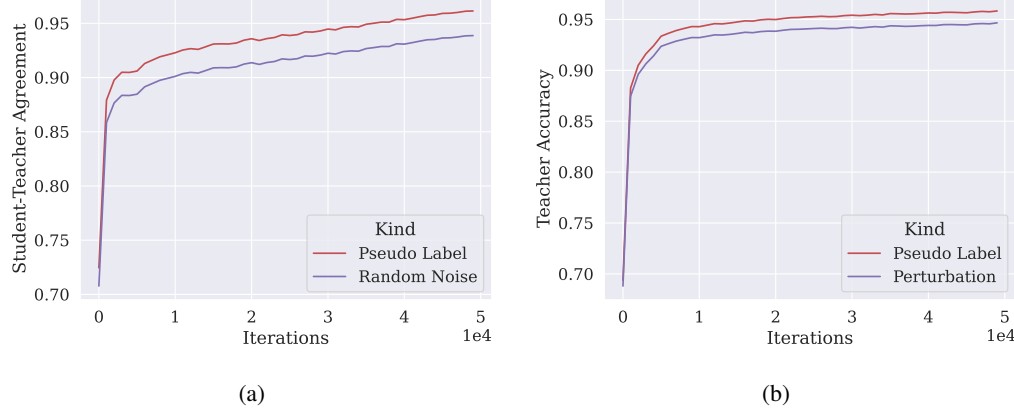

(a)                                                          (b)

Figure 11: Agreement between the student and for the pseudo label and the pseudo label with random noise (left) and the accuracy of the pseudo labels and their perturbations (right). In general both the agreement and the accuracy of the pseudo labels is high. In the case of the pseudo labels (left) adding random noise will understandably reduce the agreement between student and teacher by the amount of added noise. Considering the accuracy of the teacher's predictions, output perturbations in general slightly reduce the accuracy.

perturbation. Still, the difference between the student's and teacher's predictions reduces to less than $4\%$ of the input. This number will be further reduced with common confidence thresholding approaches, which remove low confidence predictions - which represent a likely source of the differences. In this setting, the training signal is small and will originate from a tiny subset of the input.

Adding noise to the teacher, although unlikely to improve the accuracy of the teacher, widens the gap between student and teacher. We hypothesize that this added noise could help the student to escape from unfavorable local minima or saddle points that the model can get stuck in when concentrating on the few high confidence predictions that produce a training signal.

Similarly, now examining the accuracy of the perturbations in SOP, we see that there persists a delta between pseudo-label and perturbation. Figure 5 has quantified the difference introduced with SOP, and Figure 11b shows that this difference does come at the cost of a slight accuracy decrease. Even though the perturbed pseudo-labels are less accurate, the newly introduced variety overall helps the semi-supervised approach. Close to our theory regarding the random noise, our working hypothesis is that the clearer structure, which relates to the predicted evidence from the current sample, more effectively pushes the student from locked-in positions.

