# OpenReview forum: "Sinkhorn Output Perturbations: Structured Pseudo-Label Noise in Semi-Supervised Segmentation"
_ICLR.cc/2024/Conference — Submitted to ICLR 2024_

### Official Review · Reviewer_151F · 2023-10-29

**Soundness:** 2 fair
**Presentation:** 2 fair
**Contribution:** 2 fair
**Rating:** 5
**Confidence:** 5

**Summary:**

This paper presents an approach for performing semi-supervised semantic image segmentation. The method utilizes Sinkhorn Output Perturbations in the prediction space to further utilize and improve the quality of pseudo labels. The author pointed out that there are few prior works that have explored perturbation strategies for the prediction space, and thus proposes a new perturbation method, dubbed Sinkhorn Output Perturbations.

**Strengths:**

The paper is easy to follow and well-structured, altough there are some typos. This paper proposes an interesting research topic, make a perturbed prediction based on suggested softmax-mining, beta interpolation and input patching. The results show somehow improved performance in some experiment settings.

**Weaknesses:**

-Lack of justification for the perturbation method.
    Many perturbation methods widely used in semi-supervised learning (e.g., cut-mix) have strong motivation and have been proven for their usage; they can create powerful yet effective augmented images that can leverage unlabeled data. However, the proposed perturbation method, especially in the output space, lacks a concrete rationale for its usage. Why is this type of perturbation needed? As described in the paper, the author claims that it can guide the parameter update away from amplifying the current signal. However, as also described in the method section, the augmentation method strongly depends on the "current status" of the model, which leads to contradictory arguments.

-Lack of performance improvement.
    The output of unlabeled data is augmented by the proposed method and then used as pseudo-labels for self-training. However, the proposed method fails to demonstrate its effectiveness in the PASCAL-VOC setting. It cannot achieve state-of-the-art performance in at least one experimental setting (Table 2). It does achieve comparable results in the Cityscapes setting but fails to exhibit its versatility in the PASCAL-VOC setting.

- Lack of additional experiments.
    As mentioned earlier, the ultimate goal of the perturbation method is to generate another version of pseudo-labels from unlabeled data. Are the generated augmented pseudo-labels more accurate than the original ones? The author should have presented their quantitative results.

**Questions:**

Please address my concerns. And there are few typos in the paper (e.g., citation in the introduction section).

---

> ### Author Response · Authors · 2023-11-20
> **Response to 151F**
>
> We thank the reviewer for the careful analysis of our work. We have corrected the typos in the citations and checked the entire text again for such typos. The weaknesses identified are addressed below:
>
> ## Addressing Weaknesses
>
> ### 1a - Justification for Output Perturbations:
> Input perturbations must be carefully chosen, and their results generally do not scale with the strength of the perturbations -  i.e there is a limit to the amount of distortion, afterwards the performance deteriorates. This is particularly true in segmentation, where a direct pixel-to-pixel mapping between student and teacher is required when trained with pixel-wise losses. We have added evidence to the appendix showing that even with cut-mix augmentations, the difference between student and teacher diminishes considerably during training.
> Output perturbations, however, can increase the differences between models and ensure a persistent training signal.
>
> ### 1b - Dependency on the models state
> While the reviewer is correct that we depend on the state of the teacher in our approach, this limitation is inherent to student-teacher methodologies. Our goal is to decrease this dependency, which we empirically demonstrate with the performance of output perturbations. We do not contradict this goal by devising the softmax distribution, since although this is absolutely based on the state of the teacher model, we are capturing outcomes that are hidden to competing methods.
>
> ### 2a - Performance Improvements on Pascal VOC
> The reviewer correctly notes that SOP *alone* does not achieve state-of-the-art results in any Pascal VOC setting. As highlighted by the reviewer, cut-mix is a vital input perturbation scheme that all algorithms in Tables 1 and 2 (including SOP) either use directly (UniMatch, CPS, .. ) or specialize to their use case (AEL, AugSeg).
> However, cut-mix by itself does not achieve state-of-the-art results. In Table 3 and Section 4.3, we discuss the effects of adding SOP to UniMatch as a state-of-the-art model on Pascal VOC. Table 3 presents findings for the partitions where the difference between SOP and the state-of-the-art method was largest, and Figure 8 in Appendix A.4 shows the validation performance throughout training.
> On Cityscapes, SOP achieves very competitive results even without the perturbation scheme used in UniMatch. We hypothesize that since Cityscapes consists of more low-frequency classes than Pascal VOC, small changes to the full pseudo-labels can have significant effects, especially for low-frequency classes, as seen in Figure 3. In contrast, Pascal VOC, with fewer low-frequency classes, sees a lesser effect from the lower amount of changed labels.
> To conclude and shown in our paper, SOP can lead to further improvements *when combined* with a state-of-the-art approach that utilizes strong input perturbations (like UniMatch).
>
> ### 2b - Additional Experiments and Accuracy of Perturbations:
> We appreciate the reviewer's suggestion and have included general accuracy figures in the appendix. Additionally, we want to reiterate that we do not present SOP as a refinement scheme that strictly leads to more accurate pseudo-labels. SOP aims to introduce variety, in a more structured manner compared to just random noise. We believe that our empirical results support this case.
> In Appendix A.3, we provide a quantitative analysis and a qualitative description of SOP with accurate distribution targets. We study SOP with a distribution oracle, where the targets for the Sinkhorn algorithm are obtained from the ground truth. While having access to the ground truth distribution is unrealistic in practice, this provides insight into our allocation algorithm's performs with more accurate distribution estimates.
>
> We again thank the reviewer for the thorough analysis of our work and for highlighting the missing context of the accuracy of the perturbations.
> We hope that that our responses and the additions to the text have improved the reviewer's assessment of our work.

---

> > ### Comment · Reviewer_151F · 2023-11-22
> >
> > Thank you for the author's response. The reviewer has carefully read the response and agrees that it has addressed some concerns. However, major issues still exist: Effectiveness of the Proposed Method: The reviewer notes that the SOP alone cannot achieve state-of-the-art performance, especially for PASCAL-VOC, and relies on unimatch, one of the effective perturbation methods in the literature. The reviewer is concerned about the proposed method's effectiveness and questions why it does not work well for PASCAL-VOC but performs well for Cityscapes. Is the method versatile enough Pseudo Labels Inaccuracy and Improved Performance: 2) Basically, the SOP makes pseudo labels more inaccurate, as the author responded. But the model trained with them improves its performance. How this phenomena happened? The author should have suggest (at least) a simple theory and some experiments to prove it experimentally. Based on this observation, I'll keep my rating as 5, marginally below the acceptance threshold.

---

### Official Review · Reviewer_dzt9 · 2023-11-02

**Soundness:** 3 good
**Presentation:** 2 fair
**Contribution:** 2 fair
**Rating:** 6
**Confidence:** 3

**Summary:**

This paper targets semi-supervised semantic segmentation. It proposes Sinkhorn Output Perturbations, which adds structured pseudo label noise to the training procedure. The proposed method can be seamlessly plugged into existing methods. Experiment results prove the effectiveness of the proposed method.

**Strengths:**

1. The proposed method is orthogonal to existing methods.
2. Overall the paper is well-written and easy to follow.

**Weaknesses:**

1. The related work section is a little bit concise. You can have more discussions on related works in revision.
2. The novelty of the proposed method is a little bit limited to me, though the performance is convincing and the method is practical. Could you have more explanations on the novelty of your method?

**Questions:**

See weakness.

---

> ### Author Response · Authors · 2023-11-20
> **Response to dzt9**
>
> We thank the reviewer for the kind review and highlighting the strengths of our work.
>
>
> ### 1 - Related Work Section:
> We understand the reviewer's comment regarding the conciseness of the related work section. However, we are constrained by space limitations. We have also been requested by other reviewers to add empirical data, which further limits our ability to expand this section. We acknowledge the importance of discussing related work and have endeavored to balance this with the other content requirements within the allowed space.
> We have added two additional refrences (1 survey) and hope that the section now gives the reader the adequate means to come to up to speed.
>
> ### 2 - Novelty of the Proposed Method:
> We appreciate the reviewer's recognition of our method's practicality and performance. To further clarify the novelty:
> - Our primary contribution is the application of optimal transport to segmentations with an estimated distribution as guiding target. This represents a novel approach in the context of semi-supervised learning. The scheme is orthogonal to common semi-supervised segmentation approaches and can be added on top of other methods. Moreover, and see in Appendix A.3, our approach could be used as a weakly supervised algorithm when accurate or only general information about the type and size of the visible objects is available.
> - We have extended the strong-weak perturbation scheme in semi-supervised learning to include model outputs. Unlike previous works that focused on output refinement, our strategy specifically increases the noise floor in the student-teacher optimization process with predictions that are not direct results of the teacher's output.

---

### Official Review · Reviewer_XPw6 · 2023-11-02

**Soundness:** 2 fair
**Presentation:** 3 good
**Contribution:** 2 fair
**Rating:** 6
**Confidence:** 3

**Summary:**

This paper studies the teacher-student scheme in semi-supervised segmentation. It goes beyond input and feature perturbations and introduces Sinkhorn Output Perturbations to add structured pseudo-label noise to the training. Sinkhorn output perturbations can serve as a plugin to be integrated into any segmentation model. The proposed method achieves SoTA on Cityscapes and competitive performance on Pascal VOC 2012.

**Strengths:**

1. Most papers consider perturbations in inputs and feature levels. This paper introduces output-level perturbations which involve the structural noise of an image. From this perspective, it is novel to some extent.

2. The writing is easy to follow. The motivation is clear.

**Weaknesses:**

1. The performance is less strong to prove the effectiveness of the proposed method, especially on Pascal VOC 2012.

2. For the idea of introducing global class distribution to perturb local per-pixel predictions, are the local predictions really influenced by the global class (per-image level) distributions? If so, how about maintaining the class relationships based on the whole dataset (i.e., calculate P_{sm} in Eq. 5 based on the whole training set)?

**Questions:**

Please see Weakness.

---

> ### Author Response · Authors · 2023-11-20
> **Response to XPw6**
>
> We thank the reviewer for recognizing the novelty of our work and for the insightful questions.
>
> ### 1 - Performance on Different Datasets:
>
> The reviewer rightly points out that our perturbation scheme **alone** does not achieve state-of-the-art results on both datasets. However, when SOP is combined with existing state-of-the-art semi-supervised algorithms, such as UniMatch, the results are notably improved. This is detailed in Table 3 and Section 4.3.
> Pascal VOC, with its less dense labeling compared to Cityscapes, presents a different challenge. SOP's impact is more pronounced in low-frequency classes, as shown in Figure 3. However, due to Pascal VOC's typically larger area per class, this effect is somewhat diminished.
> Our work demonstrates that output perturbations, when **added** to strong input perturbations (as shown in related work like AugSeg  and UniMatch), can further enhance model performance on Pascal VOC (Section 4.3 and Table 3).
>
> ### 2a - Influence of Global Class Distribution:
>
> We provide three types of evidence to support the impact of global class distribution on local predictions:
> - Figure 5 showcases the average number of labels changed by our perturbation scheme, so in general there is change happening in every iteration.
> - Table 4 and Figure 2 highlight the effects of decreasing window size for the distribution input and show the effects of more localized information.
> - Additionally, in A.3, we demonstrate the results of perturbing with a distribution oracle, a weakly supervised approach where the true distribution is known. The results demonstrate that the global distribution has an effect on a pixel level, since SOP with the distribution oracle further increases the mIoU. Figure 7 also shows some reallocations by the Sinkhorn algorithm and the true distribution and demonstrates a potential upper bound of the approach.
>
> ### 2b - Regarding Dataset-Wide Distribution:
> We believe that a general per-dataset distribution is less useful in segmentation tasks compared to classification (SLA [1]). In contrast to classification, in segmentation input perturbations such as resizing and cropping will alter the label distribution per image, making a training set-based estimated data distribution unreliable.
> Furthermore, the computational load for global reallocation is significant. Our batched algorithm (Appendix A.1) calculates Sinkhorn iterations independently for each image in the batch. In contrast, a global approach would require a single computation for a massive cost matrix ( 1 x HWN x C, where N is the number of images), rapidly exhausting GPU memory even for small datasets.
>
>
> [1] Kai Sheng Tai, Peter Bailis, and Gregory Valiant. Sinkhorn Label Allocation: Semi-supervised
> classification via annealed self-training. In International Conference on Machine Learning, 2021.

---

> > ### Comment · Reviewer_XPw6 · 2023-11-22
> >
> > Thank the authors for their response. I confirm my positive rating for this submission.

---

### Official Review · Reviewer_tvp8 · 2023-11-02

**Soundness:** 2 fair
**Presentation:** 2 fair
**Contribution:** 2 fair
**Rating:** 5
**Confidence:** 3

**Summary:**

1. For the semi-supervised segmentation task, this paper extends the previous strong-weak augmentation methods. Instead of solely augmenting the input or features, this approach augments the output of the teacher model.
2. The proposed method constructs plausible pseudo-labels by randomly interpolating the one-hot and softmax class distributions. This paper thinks it can soften the inherent limitations of student-teacher methodologies.
3. The experimental results demonstrate that the proposed approach achieves state-of-the-art (SOTA) results in Cityscapes and slightly lower than SOTA results in Pascal VOC 2012. Additionally, the experiments show that this method can further enhance the performance of Uni-Match.

**Strengths:**

1. The authors propose perturbing output instead of input or features, which is an interesting problem.
2. In the experimental results, their method surpasses other semi-supervised segmentation methods on the Cityscapes dataset, particularly showing significant gains with little data (76.0% vs 75.0%).

**Weaknesses:**

1. The lack of insight and direct evidence for why perturbing the original teacher output can improve model performance, and why adding random noise can also be effective. The authors claim that it can cover possible correct predictions, but they do not provide specific numbers or charts for detailed explanation.

2. Additionally, the description of the ablation experiments is not clear enough, as it does not provide a direct comparison between the results of no output perturbation, random noise, and SOP. As a result, it is hard to understand the actual impact of SOP relative to the model without output perturbation or with random noise.

**Questions:**

1. The authors could provide more explanations, either theoretically or experimentally, regarding why adding noise to the output can improve semi supervised model performance.
2. The authors could explain the reason behind interpolating between one-hot and softmax distribution and why this approach can be effective.


Update

Thanks to the author‘s response. It proposes some new hypotheses and insights (hypothesize that this added noise could help the student to escape from unfavorable local minima or saddle points), solving some of the concerns. I still think that this paper may be able to provide more solid evidence and theory for output perturbation, as well as a direct and clear ablation comparison for sop.

---

> ### Author Response · Authors · 2023-11-20
> **Response tvp8**
>
> We appreciate the feedback provided by the reviewer and the opportunity to clarify aspects of our research. Below, we address the highlighted weaknesses and questions.
>
> ### W1 - Effects of Noise and Perturbation
> We have tried to use careful language in order not to imply that the main mechanism of SOP is improving the accuracy of the pseudo-labels and thus improving the quality of the teacher.
> SOP is **not** a prediction refinement mechanism and will, in fact, generally lower the accuracy of the pseudo-labels. (We have added a section to the appendix)
> The main mechanism of SOP is that it introduces variety that is not necessarily achievable by either just following or refining the teacher.
> This behavior is observable in Figure 3, which also highlights a potential reason why the results on Cityscapes benefited more from our perturbation than on Pascal VOC.
>
> Even though SOP, in general, will only reallocate between 1.5 to 2.5% of all labels, Fig 3b,c shows that for low-frequency classes this overall small increase for all classes can lead to large changes for the underrepresented ones.
> In the shown example, some classes have grown or been completely removed. Although there may be cases where the perturbations will be more accurate, the main effect will be pushing the model off the learning trajectory if it was only following the teacher's pseudo-labels, by the introduction of these alternative views of the scene.
> In that sense, it is our theory that the added structural noise can potentially help the model to move out of saddle points or local minima, and have similar effects as the gradient noise in mini-batch SGD.
>
> Apart from the evidence in the text, we have added sections to the appendix that analyze the difference between the student and teacher's predictions with and without random noise, as well as an evaluation of the accuracy of the pseudo-labels and their perturbations during training. The results show that, in general, the perturbations are less accurate than just the pseudo-labels, and this is reflected in Figure 6 (left).
> Increasing the weight of the perturbations above a threshold will hamper the performance of the model, so there is a balance between exploration and deterioration.
>
> ### W2- Description of the Ablations:
> We thank the reviewer for highlighting the subpar description of the baseline. We do cover the baseline in the paper, shown in Figure 4, and we have added the numerical values for the Pascal VOC baseline in section 4.1 in the text. The results given in the Figure and section for 0% label noise represent the semi-supervised baseline, as the results originate from the semi-supervised student-teacher model that is the core model for all our experiments, without perturbed or randomized targets.
> We have chosen to omit these results from the main Tables 1 and 2, in order to focus on the comparison with other semi-supervised methods.
>
>
> ### Q1 - Additional Experiments
> We have added further sections to the appendix that discuss further relevant properties of training with either random noise or SOP. See also our comments W1 above.
>
> ### Q2- Interpolation Between One-Hot and Softmax Distributions
> Only targeting the one-hot distribution with the Sinkhorn algorithm would mean that this is just reflecting the present state of the pseudo-labels. It is possible that low-cost areas could be relabeled without changing the overall distribution, but since the difference between the distribution of the perturbed segmentation and the one-hot distribution would be minimal from the start, the algorithm (Appendix A.1) would quickly terminate.
>
> Only targeting the softmax distribution could shift the label distribution too far from the high-confidence prediction. In Figure 1, we do depict the one-hot and softmax distributions. Just following the latter one would massively overinflate (in this example) the low-frequency classes and produce unrealistic perturbations.
>
> We discuss the need to balance both sides in Section 3.2.1, but the essential idea is that by interpolating between the two, with a strong bias to stay close to the one-hot distribution, we can introduce some of the softmax distributions, which will accumulate evidence of classes with elevated probability and thus when used with the batched Sinkhorn algorithm, can produce overlooked objects.
> Starting from the one-hot distribution and interpolating either toward or away from the softmax distribution can emphasize classes/objects that the model deemed as probable or, in the case of interpolating away, further strengthen the evidence of the most certain classes.
>
> It can be thought of as a temperature softmax on a global level - sharpening or smoothing the present distribution.

---

### Meta-Review · Area_Chair_PV2k · 2023-12-14

**Metareview:**

The paper proposes a procedure for perturbing the pseudo-labels for semi-supervised semantic segmentation. It shows that such perturbation achieves state-of-the-art results on CityScapes and comparable results on Pascal VOC 2012. The distinction in the performance on these two main benchmarks is attributed to the larger class imbalance present in CityScapes.


All reviewers read the rebuttal and have eventually rated the paper as borderline. The main criticisms of the paper, as raised by the reviewers, is with regards to the contribution. In case the paper is to be taken as an incremental empirical work, it does not achieve state-of-the-art results consistently and if, instead, the paper is proposing a new concept, there is no formal discussion nor ablation studies that can makes the reader understand the possible reasons for effectiveness.


The AC agrees that the contribution should become empirically and/or theoretically stronger for acceptance. Therefore, the AC suggests rejection.


For the next version of the paper,
(1) it is good to cite and compare with similar methods for noisy label learning: e.g. [a].
(2) Since the improvement comes from the class imbalance present in Cityscapes, it is important to include other works that are tackling class imbalance in semi-supervised semantic segmentation, e.g. [b]


[a] NOISE AGAINST NOISE: STOCHASTIC LABEL NOISE
HELPS COMBAT INHERENT LABEL NOISE, ICLR 2021
[b] Unbiased Subclass Regularization for Semi-Supervised Semantic Segmentation CVPR 2022

**Justification For Why Not Higher Score:**

The paper's contribution is not clear. Label perturbation for robustness to noise has been explored before and class imbalance has been studied to improve semi-supervised semantic segmentation. Therefore, the paper is not ready for publication.

**Justification For Why Not Lower Score:**

N/A

---

### Decision · Program_Chairs · 2024-01-16

Reject